

# Multiple Andreev reflections in diffusive SINIS and SIFIS junctions

Artem V. Polkin[1,2] and Pavel A. Ioselevich[2,3,4⋆]

**1** National Research University Higher School of Economics, 101000 Moscow, Russia
**2** L. D. Landau Institute for Theoretical Physics, Kosygin str. 2, 119334 Moscow, Russia
**3** Department of Condensed Matter Physics,
Weizmann Institute of Science, Rehovot 76100, Israel
**4** Russian Quantum Center, Skolkovo innovation city, 121205 Moscow, Russia

⋆ pioselevich@itp.ac.ru

## Abstract

We study Multiple Andreev Reflections in long diffusive superconductor(S)-normal metal(N)-superconductor junctions with low-transparency interfaces. Assuming strong thermalization in the weak link, we calculate the current-voltage dependence $I(V)$. At intermediate temperatures, $\varepsilon_{\mathrm{Th}} \ll T \ll \Delta$, the current is dominated by noncoherent multiple Andreev reflections and is obtained analytically. The results are generalized to a ferromagnetic junction. We find that the exchange field produces a non-trivial splitting of the subharmonic gap structure. This effect relies on thermalization and vanishes in SFS junctions with no energy relaxation in the weak link.



# 1 Introduction

Andreev reflection (AR) is the process of an electron reflecting off a superconductor as a hole while the superconducting condensate gains an extra Cooper pair [1]. This basic mechanism underlies many phenomena observed in superconducting heterostructures. In particular, it helps understand the proximity effect – superconducting behaviour observed in normal metals in contact with superconductors. The Josephson effect is a prime example: electrons within the normal region of an SNS junction experience Andreev reflection at the NS interfaces, while going back and forth between the two NS interfaces. In a stationary setup such a scattering state forms an Andreev bound state which shuttles Cooper pairs between the leads, carrying a supercurrent across the junction. That is the stationary Josephson effect.

Multiple Andreev Reflections (MAR) is the mechanism behind the subharmonic gap structure (SGS) of current-voltage characteristic (CVC) $I(V)$ of a biased SNS junction [2, 3]. At voltages below the superconducting gap $2\Delta$, electrons that enter the normal region from the valence band of the left superconductor (at voltage $V$) do not have enough energy to enter the conductance band of the right superconductor. However, once the electron has experienced two Andreev reflections, coming full circle, it will have transported a Cooper pair between the leads. The pair energy difference $2eV$ is accumulated by the electron. After a number of iterations, enough energy will build up to enter the conductance band of one of the leads as schematically shown on Fig. 1. The neccessary number of Andreev reflections changes by one every time $eV$ passes through $2\Delta/n$, leading to SGS in $I(V)$.

While the idea of MAR is relatively simple, calculation of the current in real systems proves complicated. The sequence of Andreev reflections at alternating NS interfaces outlined above only works in a ballistic link with transparent NS interfaces. This was precisely the model initially proposed in Ref. [4]. Normal scattering mixes up this simple picture and produces complicated interference between different trajectories. This is furthermore complicated by the time dependence of the Andreev reflection amplitude $r_A \propto \exp i\varphi(t)$ where $\varphi(t)$ is the

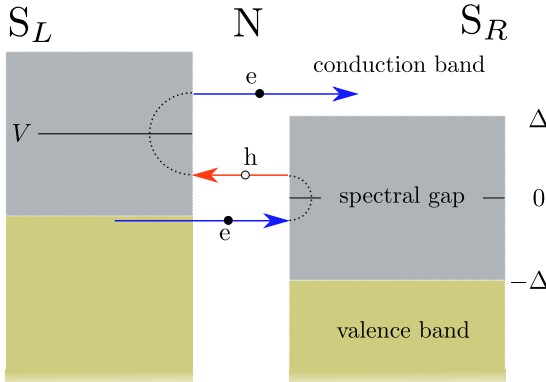

Figure 1: Semiconductor picture of MAR-assisted transport. Blue lines represent electrons, red lines represent holes, black dotted lines represent acts of AR. An electron from the valence band of $S_L$ enters the normal region where it builds up energy via AR, ultimately escaping into the conductance band of $S_R$.

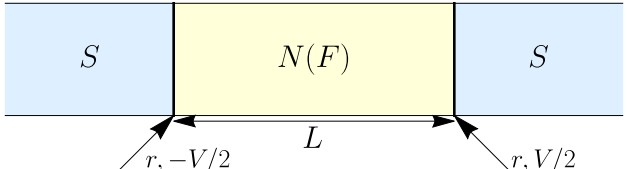

Figure 2: Schematic of junctions. We assume that all of the voltage bias $\pm V/2$ is concentrated at the SN-interfaces. The total length of the normal (ferromagnetic) region is $L \gg \sqrt{\hbar D/\Delta}$, where $\Delta$ is the order parameter of the superconducting leads and $D$ is the diffusion constant of the N(F) region. Resistance of the boundaries $R_{\text{SN}}$ is much greater than resistance of N(F)-region $R_{\text{N}}$ ($r \equiv R_{\text{SN}}/R_{\text{N}} \gg 1$).

superconducting phase. On the other hand, for a diffusive weak link with strong disorder (the so-called dirty limit, $\tau_{\text{imp}}\Delta \ll 1$), one can take advantage of Usadel equations [5] to describe the disorder-averaged behavior of the system. The proximity effect penetrates N up to the coherence length $\xi = \sqrt{\hbar D/\varepsilon}$ with diffusion constant $D$. Therefore in long junctions with $L \gg \sqrt{\hbar D/\Delta}$ MAR is incoherent. The SGS in this limit has been calculated in Ref. [6]. In short junctions MAR is coherent and has been observed [7] and studied semi-numerically [8].

All the above cases imply the absence of inelastic scattering. This is essential to the equivalent circuit method developed in Ref. [6] which relies on the conservation of energy of a quasiparticle in between Andreev reflection events. The presence of inelastic events adds another layer of complexity to the problem. Ref. [9] analytically studied SINIS junctions with strong thermalization focusing on high temperature $k_B T \gg \varepsilon_{\text{Th}}$ and low voltages $eV \lesssim \varepsilon_{\text{Th}}$, where $\varepsilon_{\text{Th}} \equiv \hbar D/L^2$ is the Thouless energy.

In recent years MAR in Josephson junctions with exotic weak links have been studied such as topological materials [10–12] or graphene [13]. In Ref. [14] SGS has been observed in an S(N/F)S junction where the weak link is a bilayer of normal metal (N) and ferromagnetic (F). Such a bilayer effectively acts as a ferromagnetic link with a diluted exchange field [15]. The measured $dI/dV(V)$ curve exhibits a double peak near a certain subgap voltage. The peaks would merge if the ferromagnetic was demagnetized and split again once the ferromagnetic was in a polarized, single-domain state. This SGS is thus sensitive to exchange field in the weak link. So far, there has been no adequate explanation of this measurement which motivates our present work.

In this work we focus on MAR in long diffusive SINIS and SIFIS junctions, as presented on Fig. 2. We assume strong thermalization in the weak link via interaction with the substrate which seems a reasonable approximation of experiment Ref. [14]. The energy relaxation only needs to be strong relative to the transport processes through the tunneling barriers (I). In this case the distribution function is close to thermal justifying the use of $\tau$-approximation to describe inelastic processes. Treating the tunneling conductance as a small parameter we construct a perturbation theory, where higher orders naturally correspond to higher numbers of Andreev reflections. We also consider the effects of an exchange field in the limit of weak energy relaxation [6] and compare results with experiment.

This paper is organized as follows. Section 2 establishes the system and its properties and introduces the Keldysh Green's function framework we use. In Sec. 3 we compute Green's function in the weak link and calculate total current through the junction. In Sec. 4 we generalize our theory to ferromagnetic junctions. In Sec. 5 we discuss the results and in Sec. 5 we conclude the paper. Details on computation of the effective temperature and the electric potential are presented in Appendices A, B, respectively. Appendices C, D contain explicit expressions related to the distribution function and the current, respectively.

## 2 Model

The system consists of a normal metal link with length $L$ much greater than $\sqrt{D/\Delta}$ (here and below we adopt units $\hbar = e = k_B = 1$) between two voltage-biased superconducting leads. We assume a symmetric junction, i.e. the SIN-interfaces have the same resistance $R_{SN} = r R_N$ where $R_N$ is the resistance of the normal region and $r \gg 1$. In order to resolve an SGS the lead temperature $T_S$ has to be much smaller than $\Delta$. We also require $T_S \gg \varepsilon_{Th}$. This will allow us to neglect electric potential effects within the weak link. In addition this suppresses coherent MAR, leaving only noncoherent MAR contributions in the current. We also assume strong electron-phonon interaction with the substrate, with a short inelastic scattering time, $\tau_{in}\varepsilon_{Th} \ll r^2$. This will allow us to calculate the distribution function in the weak link perturbatively, with a thermal Fermi distribution as a starting point. For details see Appendices A,C.

To describe the system microscopically, we follow Ref. [9], using Usadel equation on disorder-averaged semiclassical Green's function $\check{G}(t_1, t_2, \mathbf{r})$, which is a matrix in Keldysh space with components $\check{G}_{11} = \hat{G}^R$, $\check{G}_{22} = \hat{G}^A$, $\check{G}_{12} = \hat{G}^K$, $\check{G}_{21} = 0$. Here $\hat{G}^i$ are themselves matrices in particle-hole space. In mixed representation ($t = \frac{t_1+t_2}{2}$; $\tau = t_1 - t_2$), the Usadel equation takes the following form in the normal region ($x$ is measured in units of $L$).

$$-\varepsilon_{Th}\partial_x\left[\check{G} \circ \partial_x \check{G}\right] - i\varepsilon\left[\check{\sigma}_3, \check{G}\right] + \frac{1}{2}\partial_T\left\{\check{\sigma}_3, \check{G}\right\} + i\varphi_-\check{G} = \check{I}^{St}. \tag{1}$$

The $\circ$ means time convolution, which after Fourier transform over $\tau$ to $\varepsilon$ takes the form $A \circ B(\varepsilon, t) = \exp\left[\frac{i}{2}\left\{\partial_t^B\partial_\varepsilon^A - \partial_t^A\partial_\varepsilon^B\right\}\right]A(\varepsilon, t)B(\varepsilon, t)$. Here $\check{\sigma}_i$ denote $\check{\sigma}_i = 1_K \otimes \hat{\tau}_i$ with Pauli matrices $\hat{\tau}_i$ acting in particle-hole space. The electric potential $\varphi_-(t_1, t_2) = \varphi(t_1) - \varphi(t_2)$ obeys the electroneutrality condition $\varphi(t) = \frac{\pi}{4}\text{Tr}\left[G^K(t, t)\right]$ [16].

Usadel equation (1) is supplemented with tunnel boundary conditions [17]:

$$\check{G} \circ \partial_x \check{G}\big|_{x=1/2} = \frac{1}{2r}\left[\check{G}\circ, \check{G}_{right}\right]\big|_{x=1/2}, \tag{2a}$$

$$-\check{G} \circ \partial_x \check{G}\big|_{x=-1/2} = \frac{1}{2r}\left[\check{G}\circ, \check{G}_{left}\right]\big|_{x=-1/2}. \tag{2b}$$

We parametrize the Keldysh component $\hat{G}^K$ via matrix distribution function $\hat{h}$:

$$\hat{G}^{R(A)} = \begin{pmatrix} g^{R(A)}(\varepsilon) & f^{R(A)}(\varepsilon) \\ f^{R(A)}(\varepsilon) & -g^{R(A)}(\varepsilon) \end{pmatrix}, \tag{3a}$$

$$\hat{G}^K = \hat{G}^R \circ \hat{h} - \hat{h} \circ \hat{G}^A. \tag{3b}$$

In the bulk of the superconducting leads the Green's functions $\check{G}_S$ is given by

$$\hat{h}_S(\varepsilon) = \hat{1}\tanh\left(\frac{\varepsilon}{2T_S}\right), \tag{4a}$$

$$g_S^{R(A)}(\varepsilon) = \frac{\varepsilon}{\Delta}\left(\pm\eta_S - i\xi_S\right), \tag{4b}$$

$$f_S^{R(A)}(\varepsilon) = \xi_S \pm i\eta_S, \tag{4c}$$

$$\eta_S(\varepsilon) = \frac{\Delta\,\text{sign}\,\varepsilon}{\sqrt{\varepsilon^2 - \Delta^2}}\theta(|\varepsilon| - \Delta), \tag{4d}$$

$$\xi_S(\varepsilon) = \frac{\Delta}{\sqrt{\Delta^2 - \varepsilon^2}}\theta(\Delta - |\varepsilon|). \tag{4e}$$

In addition the Green's function satisfies the normalization condition $\hat{G}^{R(A)} \circ \hat{G}^{R(A)} = 1$ and the general symmetry relation between advanced and retarded functions: $\hat{G}^A = -\hat{\tau}_3\hat{G}^{R\dagger}\hat{\tau}_3$.

We can neglect the inverse proximity effect due to the assumed low transparency of the interfaces. Therefore the pairing potential $\Delta(x)$ and Green's function $\check{G}(x)$ in the superconducting leads retain their bulk values near the $NS$-interfaces and $\check{G}_{\text{right,left}}$ in our boundary conditions can be replaced with bulk Green's function of the corresponding superconducting leads.

To account for the voltage drops at the NS interfaces in our system we perform a gauge transform (given by $\check{S}_V(t) = \exp[i\check{\sigma}_3 V t]$) on the equilibrium Green's function of Eqs. (4). As a result, the Green's functions $\check{G}_{\text{right,left}}$ of the leads at the right and left NS interface can be written as $\check{G}_{\text{right,left}} = \check{S}^\dagger_{\pm V/2}(t_1) \circ \check{G}_S \circ \check{S}_{\pm V/2}(t_2)$.

The electrical current is given by the general relation [16]

$$I(t) = \frac{\pi \sigma_N}{4} \text{Tr}\left[\hat{\tau}_3 \hat{j}^K(t,t)\right], \tag{5a}$$

$$\hat{j}^K = L^{-1} \left(\check{G} \circ \partial_x \check{G}\right)^K. \tag{5b}$$

The conductivity $\sigma_N$ takes into account both electron spin projections. Coefficient $L^{-1}$ in Eq. (5b) appears due to our use of a dimensionless variable $x$.

In this paper we solve Usadel equation (1) via perturbation theory in small parameter $r^{-1}$. The first step is to determine zeroth-order approximation of the distribution function. We assume electron-phonon interaction with the substrate to be strong enough to thermalize the normal region to some effective temperature $T_e$ which is determined via heat balance equation [18–20] (for more details see Appendix A). Therefore zeroth-order approximation of matrix distribution function is diagonal with elements $h^{(0)} = \tanh \frac{\varepsilon}{2T_e}$. For low temperatures ($T_S \ll \Delta$) and $V < 2\Delta$ one can show that the difference between electron temperature $T_e$ and lead temperature $T_S$ is exponentially small (see Eq. (A.2)). Relaxation is controlled by inelastic scattering time $\tau_{\text{in}}$. To be close to thermalization the dimensionless relaxation rate $\gamma \equiv (\tau_{\text{in}}\varepsilon_{\text{Th}})^{-1}$ should be sufficiently large $\gamma \gg r^{-2}$). Physically, this inequality means that particles spend enough time in the weak link to thermalize, which justifies our choice of zeroth-order approximation.

## 3  CVC in thermalized SINIS junction

In the normal region we following Ref. [9]'s notations, parameterizing the Green's function:

$$\hat{G}^{R(A)} = \begin{pmatrix} \pm\left(1 - g_1^{R(A)}\right) & f_1^{R(A)} \\ f_2^{R(A)} & \mp(1 - g_2^{R(A)}) \end{pmatrix}. \tag{6}$$

Normalization condition then takes form

$$g_{1,2}^R = \frac{1}{2}\left(f_{1,2}^R \circ f_{2,1}^R + g_{1,2}^R \circ g_{1,2}^R\right). \tag{7}$$

One can see, that corrections to the regular Green's functions $g_{1,2}^{R(A)}$ are of a higher order in tunnel parameter $r^{-1}$ than anomalous Green's function $f_{1,2}^{R(A)}$. Therefore, we solve Usadel equation on anomalous components, and corrections to the regular part are subsequently derived from the normalization condition.

Adopting $\tau$-approximation for collision integral $\check{I}^{\text{St}}$ and taking into account suppression of electric potential $\varphi_-$ (see Appendix B for more details), we can write down the Usadel equation

and boundary conditions for $f_{1,2}^R$ expanded up to the leading order in $r^{-1}$

$$\varepsilon_{\text{Th}}\partial_x^2 f_{1,2}^R + \left[2i\varepsilon - \tau_{\text{in}}^{-1}\right]f_{1,2}^R = 0, \tag{8a}$$

$$\left.\partial_x f_{1,2}^R\right|_{x=1/2} = \frac{1}{r}f_S^R e^{\pm iVt}, \tag{8b}$$

$$\left.\partial_x f_{1,2}^R\right|_{x=-1/2} = \frac{1}{r}f_S^R e^{\mp iVt}.$$

Linearization is valid as long as $\left|f_{1,2}^R(x,\varepsilon,t)\right| \ll 1$. This is violated at energies very close to $\Delta$. Our calculations are valid as long as $|\Delta - |\varepsilon|| \gg \varepsilon_{\text{Th}}/r^2$. Mathematically, this leads to a logarithmic divergence of integrals we calculate. This is regularized by a simple cut-off. Our results do not depend on it. However higher orders of perturbation theory are sensitive to the regularization scheme.

In what follows, Advanced components of functions are found from the general symmetry relation between $\hat{G}^R$ and $\hat{G}^A$. To calculate the current, we write down values of $f_{1,2}^{R(A)}(x,\varepsilon,t)$ in the vicinity of the right boundary

$$f_{1,2}^R(x=1/2) = u^R(\varepsilon)e^{\pm iVt} + v^R(\varepsilon)e^{\mp iVt}, \tag{9a}$$

$$f_{1,2}^A(x=1/2) = u^A(\varepsilon)e^{\pm iVt} + v^A(\varepsilon)e^{\mp iVt}, \tag{9b}$$

with auxiliary functions $u(\varepsilon), v(\varepsilon)$, which correspond to incoherent and coherent propagators:

$$u^{R(A)}(\varepsilon) = -f_S^{R(A)}(\varepsilon)u(\pm\varepsilon), \tag{10a}$$

$$v^{R(A)}(\varepsilon) = -f_S^{R(A)}(\varepsilon)v(\pm\varepsilon), \tag{10b}$$

$$u(\varepsilon) = \frac{\cos\varkappa_\varepsilon}{r\varkappa_\varepsilon\sin\varkappa_\varepsilon}, \tag{10c}$$

$$v(\varepsilon) = \frac{1}{r\varkappa_\varepsilon\sin\varkappa_\varepsilon}, \tag{10d}$$

$$\varkappa_\varepsilon^2 = \frac{2i\varepsilon}{\varepsilon_{\text{Th}}} - \frac{1}{\tau_{\text{in}}\varepsilon_{\text{Th}}}. \tag{10e}$$

These results allow us to obtain corrections of the second order in $r^{-1}$ to the regular Green's function from the normalization condition Eq. (7). In the vicinity of the right superconductor they take form

$$g_1^{R(A)}(x=1/2,\varepsilon) = \frac{1}{2}\left\{u_+^{R(A)^2} + v_-^{R(A)^2} + \left(e^{2itV} + e^{-2itV}\right)u_-^{R(A)}v_+^{R(A)}\right\}, \tag{11a}$$

$$g_2^{R(A)}(x=1/2,\varepsilon) = \frac{1}{2}\left\{u_-^{R(A)^2} + v_+^{R(A)^2} + \left(e^{2itV} + e^{-2itV}\right)u_+^{R(A)}v_-^{R(A)}\right\}. \tag{11b}$$

Here we use following shorthand notation: $\Phi_\pm = \Phi(\varepsilon\pm V/2)$, $\Phi_{\pm\pm} = \Phi(\varepsilon\pm V)$, $\Phi_{\pm\pm\pm} = \Phi(\varepsilon\pm 3V/2)$. A "+" can be identified with a process related to an electron coming into the normal region from the left lead, which has a $V/2$ higher chemical potential. Thus, the $\pm$ subscripts track energy changes and allow us to identify expressions with particular MAR scenarios.

In the second order in $r^{-1}$ the Usadel equation (8a) remains the same (corrections to the linearized Usadel equation are third-order in $r^{-1}$), and to obtain second-order approximation to the anomalous Green's functions $f_{1,2}^{R(2)}$ and $f_{1,2}^{A(2)}$, we expand boundary conditions (2) up to the second order in $r^{-1}$. Near the right boundary this expansion gives

$$\left.\partial_x f_{1,2}^{R(2)}\right|_{x=\frac{1}{2}} = -\frac{1}{2r}\left[f_{1,2}^R \circ g_{S,\mp}^R + g_{S,\pm}^R \circ f_{1,2}^R\right]\Big|_{x=\frac{1}{2}}. \tag{12}$$

Here the upper(lower) sign corresponds to $f_{1,(2)}^{R(2)}$. To formulate the left boundary condition, one should change the sign of the square bracket and change $V$ to $-V$.

After some algebra, near the right boundary we find the solution in the form:

$$f_{1,(2)}^{R(2)}\left(x = \frac{1}{2}\right) = \alpha^R e^{\mp iVt} + \beta^R e^{\pm iVt}, \tag{13a}$$

$$f_{1,(2)}^{A(2)}\left(x = \frac{1}{2}\right) = -\alpha^A e^{\mp iVt} - \beta^A e^{\pm iVt}, \tag{13b}$$

$$\alpha^{R(A)} = u^{R(A)} g_S^{R(A)} v(\pm\varepsilon) + \frac{u(\pm\varepsilon)v^{R(A)}\left[g_{S,++}^{R(A)} + g_{S,--}^{R(A)}\right]}{2}, \tag{13c}$$

$$\beta^{R(A)} = u^{R(A)} g_S^{R(A)} u(\pm\varepsilon) + \frac{v(\pm\varepsilon)v^{R(A)}\left[g_{S,++}^{R(A)} + g_{S,--}^{R(A)}\right]}{2}. \tag{13d}$$

Minus sign in the r.h.s. of the relation (13b) appears due to the definition of the regular bulk Green's function (4b).

First non-vanishing corrections to the distribution function are of the second order in $r^{-1}$. We parametrize $\hat{h} = \hat{1}h_0 + \hat{\tau}_3 h_3$, and taking traces $\text{Tr}[\hat{\tau}_3 \cdot]$, $\text{Tr}[\cdot]$, of the Usadel equation and boundary conditions. This helps separate equations on $h_0$, $h_3$ and yields:

$$\varepsilon_{\text{Th}}\partial_x^2 h_{0,3} - \left(\partial_T + \tau_{\text{in}}^{-1}\right)h_{0,3} = 0, \tag{14a}$$

$$4\partial_x h_{0,3}^{(2)}\Big|_{x=1/2} = \frac{1}{2r}\left[J_1 \mp J_2\right],$$
$$4\partial_x h_{0,3}^{(2)}\Big|_{x=1/2} = \frac{1}{2r}\left[J_2 \mp J_1\right], \tag{14b}$$

$$J_{1,2} = f_{1,2}^R \circ \left[e^{\mp iVt}\left(f_S^R \delta h_-^+ - f_S^A \delta h_+^-\right)\right] + \left[e^{\pm iVt}\left(f_S^A \delta h_-^+ - f_S^R \delta h_+^-\right)\right] \circ f_{2,1}^A, \tag{14c}$$

where $\delta h^\pm = h_{S,\pm} - h^{(0)}$ (we used a $\pm$ superscript in $\delta h^\pm$ since it only refers to the energy of the first term, while a $\pm$ subscript indicates an energy shift of the subscribed as a whole). In Eqs. (14) we once again neglected the term related to electric potential $\varphi_-$. Due to the symmetry of the boundary conditions, solutions of Eq. (14a) take form of a Fourier series with 3 components presented below (explicit expressions for the coefficients are found in Appendix C):

$$h_0^{(2)} = \sum_{n=-1}^{1} A_n^{(2)}(\varepsilon)\cos(\varkappa_{nV}x)e^{2inVt}, \tag{15a}$$

$$h_3^{(2)} = \sum_{n=-1}^{1} B_n^{(2)}(\varepsilon)\sin(\varkappa_{nV}x)e^{2inVt}, \tag{15b}$$

$$\varkappa_{nV}^2 = \frac{2inV}{\varepsilon_{\text{Th}}} - \frac{1}{\tau_{\text{in}}\varepsilon_{\text{Th}}}. \tag{15c}$$

Other harmonics in $h_{0,3}(t)$ are of higher order in $r^{-1}$.

Current, determined by Usadel equation, has a constant value across the system and can be calculated at any point. It is convenient to evaluate the expression (5b) near the right superconductor, where we can make use of boundary conditions. This trick allows us to obtain the current in the order $r^{-(n+1)}$ with Green's function only calculated up to the order $r^{-n}$. Calculated this way, the leading term in the current takes form

$$I^{(0)} = \frac{1}{4R_\Sigma}\int \mathrm{d}\varepsilon\left\{\delta h^-\left(g_{S,-}^A - g_{S,-}^R\right) - \delta h^+\left(g_{S,+}^A - g_{S,+}^R\right)\right\}. \tag{16}$$

Here $R_\Sigma = 2R_{SN} + R_N \approx 2R_{SN}$ is the resistance of the junction.

In the limit of low temperatures $T_S \ll \Delta$ this integral can be evaluated, leading to the familiar square-root voltage-current relation

$$I^{(0)} = \frac{1}{R_\Sigma} \theta(V - 2\Delta)\sqrt{V^2 - (2\Delta)^2}. \tag{17}$$

Here $\theta(x)$ is Heaviside theta function. To observe an SGS in $I(V)$ we need to go to higher order in $1/r$.

Applying the same procedure to the first order corrections to Green's function, we obtain relation for the first-order correction of the current $I(t)$.

$$I^{(1)}(t) = \frac{1}{8R_\Sigma} \left\{ J_0^{(1)} + \left[ J_{h_s}^{(1)} + J_+^{(1)} e^{2iVt} + J_-^{(1)} e^{-2iVt} \right] \right\}. \tag{18}$$

Here $J_i^{(1)}$ represent various integrals which are explicitly listed in Appendix D (except for $J_0^{(1)}$ which is given below). All terms in the square brackets depend on time, which suggests they refer to coherent MAR and should be negligible (time dependence can only emerge from a dependence on superconducting phase difference, which in turn implies coherence). This is indeed the case: all of them contain $v(\varepsilon)$ which is exponentially small at energies $\varepsilon \gg \varepsilon_{\text{Th}}$, while $J_0^{(1)}$ contains $u(\varepsilon)$ which does not contain exponential smallness. Thus, the only remaining term is

$$J_0^{(1)} \equiv \int d\varepsilon \tanh \frac{\varepsilon}{2T_e} \left[ \left( f_{S,-}^A + f_{S,-}^R \right)\left( u_-^A + u_-^R \right) - \left( f_{S,+}^A + f_{S,+}^R \right)\left( u_+^A + u_+^R \right) \right]. \tag{19}$$

It produces a relatively small subgap current, see Fig. 3, which corresponds to inelastic scattering processes in the weak link. It also enhances the current at $V \gtrsim 2\Delta$. These results, alongside with (D.2b), agree with known CVC results at subgap voltages, calculated in adiabatic approximation in Ref. [21], where superconducting correlations (i.e. anomalous Green's function $f_{1,2}^{R,A} = 0$) were neglected in the normal region. However, within our approach we cannot calculate the excessive/deficient current, since at these voltages the distribution function is no longer close to thermal (see also Appendix A).

In order to observe effects of Andreev reflections, we need to compute the second order approximation.

After employing the same scheme of calculations, we obtain the following expression for the stationary contribution to the current:

$$I^{(2)} = \frac{1}{8R_\Sigma} \left[ \frac{-1}{4\varkappa_0 r}\left( \cot \frac{\varkappa_0}{2} + \tan \frac{\varkappa_0}{2} \right) J_h^{(2)} + J_f^{(2)} + J_g^{(2)} \right]. \tag{20}$$

Objects $J_h^{(2)}, J_g^{(2)}, J_f^{(2)}$ represent rather cumbersome integrals, written explicitly in Appendix D. The three terms correspond to contributions produced from including second-order corrections to the functions $h_{0,3}, g_{1,2}^{R(A)}, f_{1,2}^{R(A)}$ respectively.

Numerical computations, presented on Fig. 4, reveal that $J_h^{(2)}$ is the term responsible for sharp features in the voltage dependence. At voltages close to $2\Delta/3$ this term exhibits square-root behavior (see Fig. 5, which is smeared for higher temperatures. Direct calculation produces the analytical result

$$J_h^{(2)}\left( V \sim \frac{2\Delta}{3} \right) = 9\sqrt{3\Delta}\sqrt{V - \frac{2}{3}\Delta}\left[ u\left( \frac{\Delta}{3} \right) + u\left( -\frac{\Delta}{3} \right) \right] \theta\left( V - \frac{2}{3}\Delta \right), \tag{21}$$

where functions $u(\varepsilon), v(\varepsilon)$ are defined in Eq. (10).

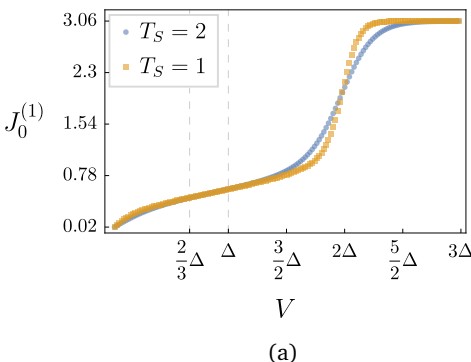
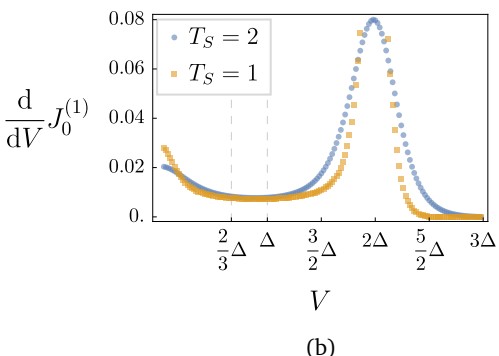

(a)                                              (b)

Figure 3: First order contribution to the current $J_0^{(1)}$ (a) and to the differential conductance $\frac{\mathrm{d}}{\mathrm{d}V}J_0^{(1)}$ (b) for different temperatures. The subgap current should be understood as a result of inelastic scattering in normal region. Here $\Delta = 30$, $\varepsilon_{\mathrm{Th}} = 1/8$, $\gamma = 0.1$, $r = 40$. Peak at $V = 2\Delta$ in the differential conductance corresponds to transition to the regular quasiparticle regime. At voltages above $2\Delta$ our results are only qualitatively correct since the close-to-thermal approximation of the distribution function breaks down.

We associate the square-root feature at $V \approx \frac{2}{3}\Delta$ with the onset of MAR transport involving two Andreev reflections.

Notice that $J_h^{(2)}$ in Eq.(20) comes with a factor that leads to exponential suppression at large $\gamma$. Expanding the first term of Eq. (20) in orders of $\gamma$ we get.

$$I_h^{(2)} \approx \frac{e^{-\sqrt{\gamma}}}{8\sqrt{\gamma}R_\Sigma r}J_h^{(2)}. \tag{22}$$

This limit corresponds to the super-thermalized limit where a particle thermalizes before it travels the length of the junction.

We would like to note here, that this way of evaluating integrals, corresponding to a total current, should be corrected for contributions of the third order of $r^{-1}$, because multiplication of BCS peculiarities produce nonlogarithmical divergence of integrand, therefore Green's function with energies $\varepsilon \approx \Delta$ should be evaluated more precisely.

## 4 CVC in thermalized SIFIS junction

We now turn to the SIFIS junction. We assume a homogenous exchange field $\mathbf{h}_{\mathrm{ex}}$ in the ferromagnetic link. Spin projection along $\mathbf{h}_{\mathrm{ex}}$ is then conserved in the system so that the two spin subbands can be considered independently.

The exchange field is incorporated into Usadel equation (1) by formally replacing [22] $\varepsilon$ with $\varepsilon \pm h_{\mathrm{ex}}$ where the sign corresponds to spin and $h_{\mathrm{ex}} = |\mathbf{h}_{\mathrm{ex}}|$ is measured in energy units. Since the exchange field is only present in the weak link (but not in the S leads), the substitution $\varepsilon \mapsto \varepsilon \pm h_{\mathrm{ex}}$ must only be made in functions pertaining to the weak link: $\varkappa_\varepsilon \mapsto \varkappa_{\varepsilon \pm h_{\mathrm{ex}}}$ and $\varepsilon \mapsto \varepsilon \pm h_{\mathrm{ex}}$ within the distribution function $h^{(0)}$. With these adjustments, all procedures of Sec. 3 are valid for the SIFIS junction. Note that in this case $\sigma_N$ in the general relation Eq. (5b) should be understood as the conductivity of the spin subband currently in consideration. The total current through the junction is then obtained by adding the currents carried by each spin projection, $I = I_\uparrow + I_\downarrow$.

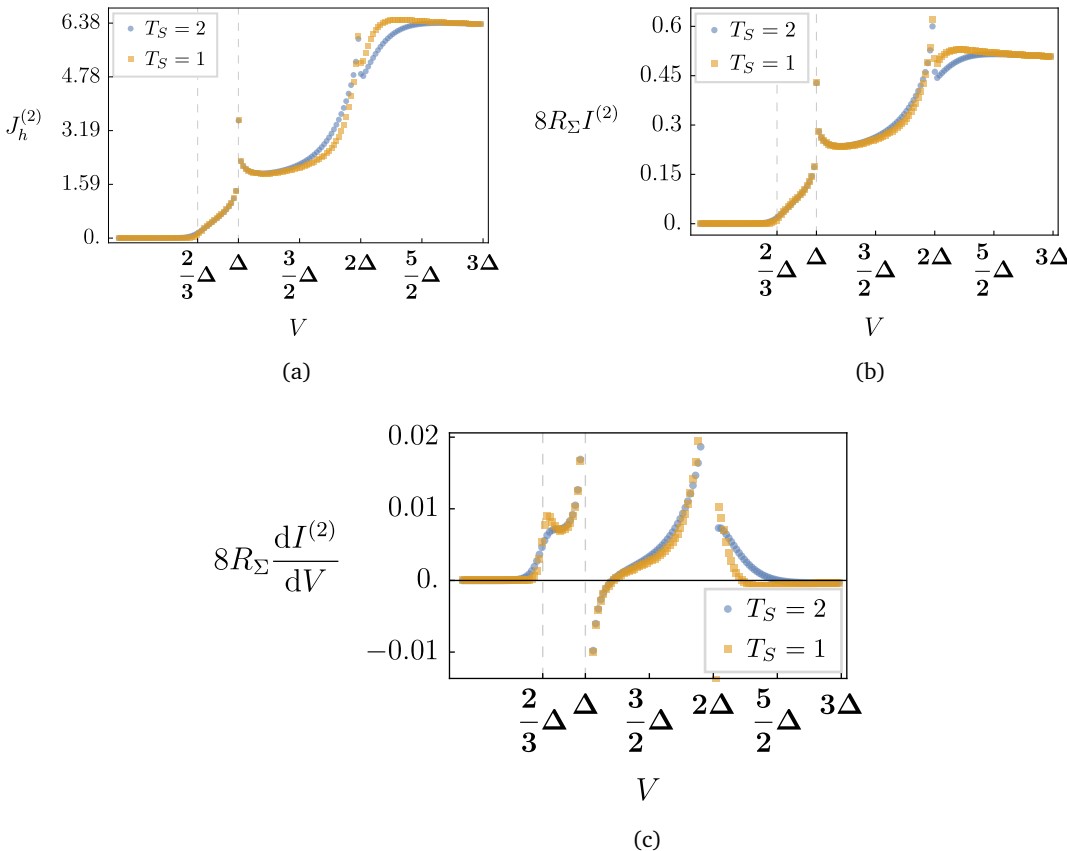

Figure 4: Numerical computation of second-order contributions to the current, $I^{(2)}$. (a) $J_h^{(2)}$, (b) $8R_\Sigma I^{(2)}$ and differential conductance (c) $8R_\Sigma \frac{\mathrm{d}}{\mathrm{d}V} I^{(2)}$. Note that terms $J_f^{(2)}, J_g^{(2)}$ of Eq. (20) do not contain MAR features and can thus be neglected compared to $I^{(1)}$ while $J_h^{(2)}$ is responsible for MAR features at $V = \Delta$ and $V = 2\Delta/3$ that were absent from the first-order term $I^{(1)}$. The plots were done at $\Delta = 30$, $\varepsilon_{\mathrm{Th}} = 1/8$, $\gamma = 0.1$, $r = 40$.

A numerical comparison of different contributions to the SIFIS current is presented on Fig. 6. The primary effect of the non-zero exchange field is the splitting of features in $J_h^{(2)}$ at $V \sim \Delta, \frac{2}{3}\Delta$. This is confirmed by low-temperature asymptotic expansions:

$$
J_h^{(2)}\left(V \approx \frac{2}{3}\Delta\right) = \frac{9\sqrt{3\Delta}}{2}\left[\sqrt{V - \frac{2}{3}(\Delta + h_{\mathrm{ex}})}\{u(\Delta/3 + h_{\mathrm{ex}}) + u(-\Delta/3 - h_{\mathrm{ex}})\} + \right.
$$
$$
\left. \sqrt{V - \frac{2}{3}(\Delta - h_{\mathrm{ex}})}\{u(\Delta/3 - h_{\mathrm{ex}}) + u(-\Delta/3 + h_{\mathrm{ex}})\}\right], \quad (23a)
$$

$$
J_h^{(2)}(V \approx \Delta) = \frac{4\Delta}{r}\left(\mathrm{Re}\left[\sqrt{\frac{i\varepsilon_{\mathrm{Th}}}{\Delta}}\log\left(\frac{\varepsilon_{\mathrm{Th}}\varkappa_{\Delta + h_{\mathrm{ex}} - V}^2}{4i\Delta}\right)\right] + \right.
$$
$$
\left. \mathrm{Re}\left[\sqrt{\frac{i\varepsilon_{\mathrm{Th}}}{\Delta}}\log\left(\frac{\varepsilon_{\mathrm{Th}}\varkappa_{\Delta - h_{\mathrm{ex}} - V}^2}{4i\Delta}\right)\right]\right), \quad (23b)
$$

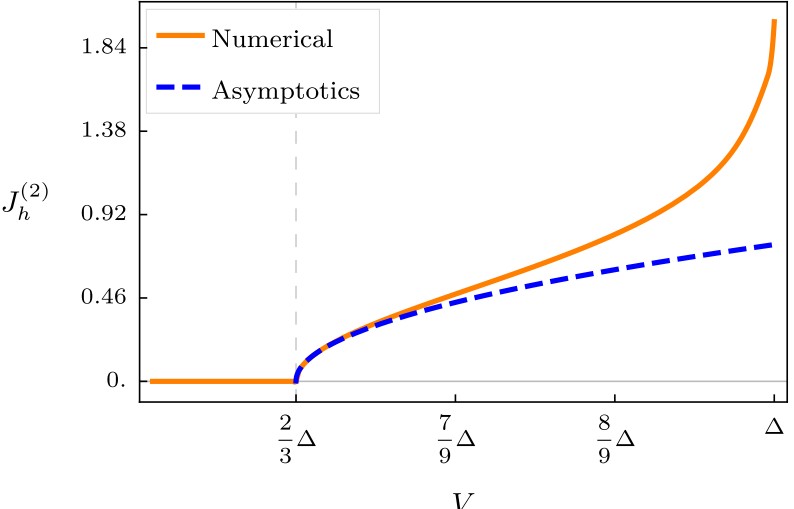

Figure 5: Comparison of low-temperature asymptotics (21) with numerical evaluation of $J_h^{(2)}$. Asymptotic expansion reveals sharp square-root growth from $V = \frac{2}{3}\Delta$. Here $\Delta = 30$, $\varepsilon_{\text{Th}} = 1/8$, $\gamma = 0.1$, $r = 40$ and $T_e = T_S = 0$.

where $u(\varepsilon)$ and $\varkappa_\varepsilon$ are defined in Eq. (10)

From relations (23) we see that the splitting is linear in $h_{\text{ex}}$ but the coefficients vary between peaks. This is somewhat expected, because the exchange field shifts energy bands as a whole. We present comparison of the results of low-temperature numerical computations via Eq. (D.2a) and asymptotic expansions (23) on Fig. 7.

## 5 Discussion

Our results for the SINIS junction agree with the established MAR rules: the SGS exhibits singularities at voltages that are fractions of $2\Delta$, i.e. $2\Delta/n$. This fits the diagram pictured on Fig. 1: we consider a particle from the valence band of $S_L$ and track its energy accumulation due to back-and-forth AR in N. Peculiarities in $I(V)$ occur whenever such a MAR ladder transports a carrier from the edge of the valence band to the edge of the conductance band. This corresponds to matching the gap $2\Delta$ with energies carried by a single electron and a number of Cooper pairs, i.e. $V + 2nV$ when travelling from one superconductor to the other or $2nV$ if the quasiparticle returns to the same lead and only Cooper pairs are transported. This produces odd and even series of MAR features in the SGS.

However, once we add exchange field to the picture and apply the same interpretation to the SIFIS case this energy-counting scheme starts contradicting our results. Suppose we transport a number of Cooper pairs across the junction. The energy released is still precisely $2V$ per Cooper pair, even with an exchange field to the weak link. The electron energy is also just $V$ and travelling through a ferromagnetic region does not change it. Therefore we must conclude that the SGS grid must remain unchanged, i.e. we still have $2\Delta/n$.

Our results Eqs.(23) indicate, however, that splitting of the SGS should happen. The short answer to this apparent paradox is that the familiar energy counting method does not work in a system with strong thermalization. In the absence of thermalization it was fair to treat the weak link as a quantum scatterer that conserves energy (or adds $nV$ to it). We attached two superconducting leads with known distribution functions to this scatterer and considered the current within the Blonder-Tinkham-Klapwijk (BTK) language [4] of Fig. 1.

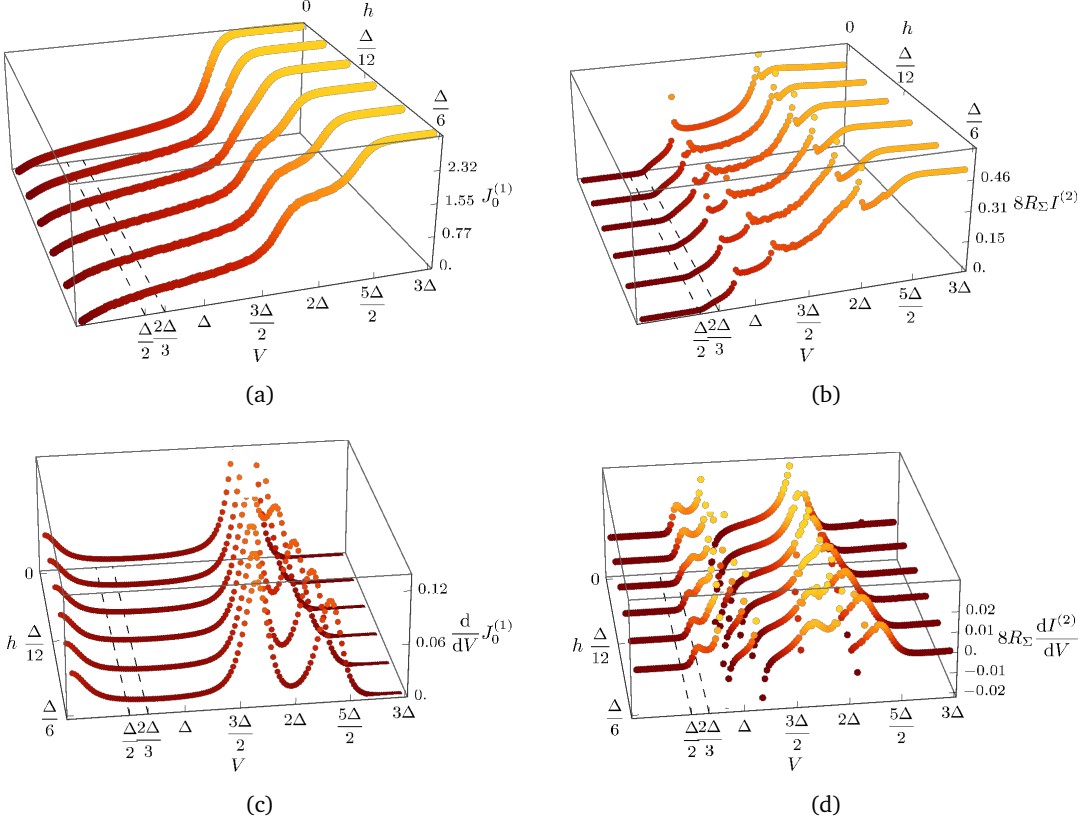

Figure 6: Numerical computations of different contribution to the current (a) $J_0^{(1)}$, (b) $8R_\Sigma I^{(2)}$ and to the differential conductance (c) $\frac{\mathrm{d}}{\mathrm{d}V}J_0^{(1)}$, (d) $8R_\Sigma \frac{\mathrm{d}}{\mathrm{d}V}I^{(2)}$. These results confirm that all peculiarities are split by exchange field $h_{\mathrm{ex}}$. Here $\Delta = 30$, $\varepsilon_{\mathrm{Th}} = 1/8$, $\gamma = 0.1$, $r = 40$.

In the strong thermalization regime considered in the present paper, the weak link should be treated as a reservoir in its own right: in the zeroth order approximation it supplies particles according to a thermal distribution function – just like a lead does. Therefore, we should not track the adventures of a quasiparticle that enters the weak link from one lead with the quest to escape into the other lead. Instead, we start with a particle that lives on the Fermi surface in N as illustrated on Fig. 8. The voltage drop between N and S is $V/2$. Thus, an electron from the Fermi surface has to accumulate $\Delta - V/2$ using the AR mechanism which provides energy in quanta of $V$, as usual. Thus, we get the SGS structure $\Delta = V(m + 1/2)$ with $m \in \mathbb{Z}$.

The introduction of an exchange field within this paradigm does split the SGS. Indeed, the distribution functions for different spins get shifted by $\pm h_{\mathrm{ex}}$. Hence, the starting energy of our charge carriers is now also shifted and we arrive at an SGS with features at $V = 2(\Delta \pm h_{\mathrm{ex}})/(2m + 1)$. This perfectly agrees with our analytical results Eq. (23).

The SGS structure in the thermalized case can also be understood from analyzing the distribution function. In the zeroth order, i.e. in the limit of disconnected leads, $r \to \infty$, electron occupation numbers in N obey a perfect Fermi distribution $h^{(0)}$. If we attach leads via tunneling junctions, dissipative current will be able to flow from N to S, provided there are electrons with $\varepsilon > \Delta - V/2$. Such electrons can be activated thermally, but this is an exponentially weak contribution. Alternatively, higher energy can be achieved via AR. Occasional AR happening at the interfaces cause a correction to the distribution function: there are now some particles within the $(0, V)$ window of energies. The amount of such particles is small in $1/r$ since it requires tunneling to occur, however unlike thermal activation there is no exponential

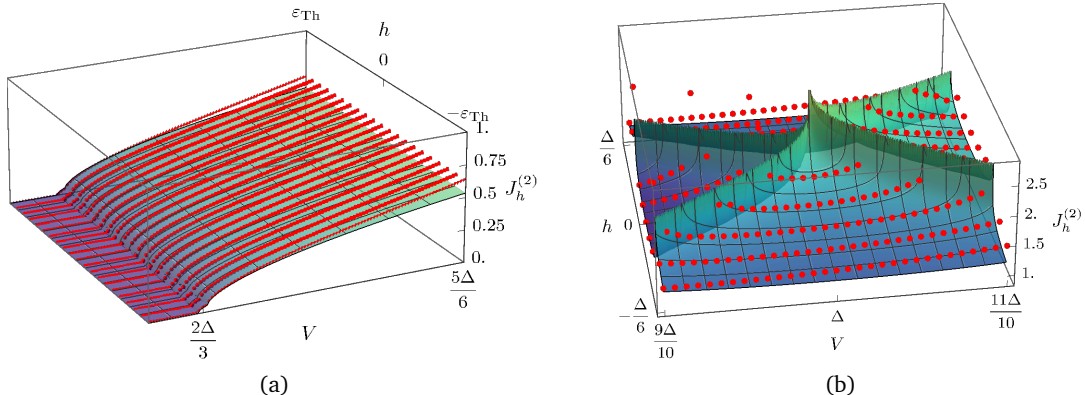

Figure 7: Comparison of contributions to a second-order current $I^{(2)}$ per spin subband, acquired with Eq. (D.2a) (red dotted lines), and asymptotic expansions of $J_h^{(2)}$ (green surface) for voltages (a) $V \approx \frac{2}{3}\Delta$, (b) $V \approx \Delta$. Here $\Delta = 30$, $\varepsilon_{\text{Th}} = 1/8$, $\gamma = 0.1$, $r = 40$, $T_e = T_S = 0$.

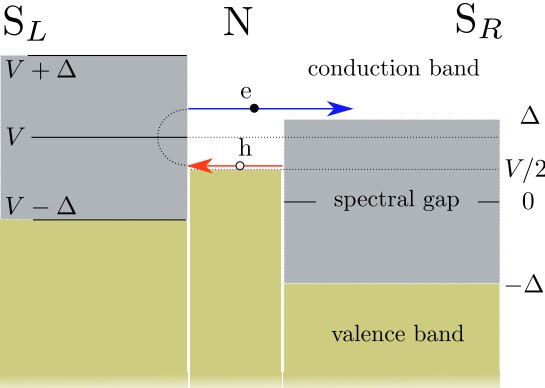

Figure 8: Semiconductor picture of MAR-assisted transport for the thermalized case. Blue lines represent electrons, red lines represent holes, black dotted lines represent acts of AR. N should be treated as a thermalized source of carriers. Particles that escape into S with the help of MAR are quickly replenished by thermalization.

smallness. Some of these particles manage to undergo another AR before energy relaxation gets them. Thus there is another window of energies, $(V, 2V)$ where occupation numbers are even smaller and given by the next order in perturbation theory in $1/r$. This MAR activation mechanism provides us with electrons with energies high enough to enter a superconductor, contributing to current. We can recognize this physics in our calculations. In Eq. (20) the term responsible for the SGS feature at $V = 2\Delta/3$ was $J_h^{(2)}$ which emerged from corrections to the distribution function $h$ caused by the tunneling boundary condition.

Note that the above picture straightforwardly produces odd SGS series. Yet our calculation also reveals features in $I(V)$ at $\Delta$, which is part of the even series. Just like the $2\Delta/3$ feature it first appears in $I^{(2)}$, i.e. in the same order of perturbation theory as the $2\Delta/3$ feature. The standard (energy-conserving) interpretation for the $V = \Delta$ signal in the SGS structure is that at this value of $V$ the following transport process becomes possible: an electron enters N from the left S, travels to the right NS interface where it experiences Andreev reflection. It then returns back into the left S as a hole, transporting a net Cooper pair through the junction. This process is only possible if the Fermi level in the right S is below the gap of the left S, i.e. $V \geq \Delta$. This scheme yields the proper order in $r$. However, it does not involve thermalization and therefore

is insensitive to exchange fields. To fix this, instead of speaking about a particular energy-conserving trajectory we must again think in terms of distribution functions. In this case we can trace back the signal at $V = \Delta$ to the interplay of corrections to the unperturbed thermalized distribution function $h^{(0)}$ from two transport processes: AR at the right NS interface, and quasiparticle transport through the left NS interface. The latter is strongest at energies just below the gap where the BCS density of states diverges. For Andreev reflection to appear, this divergence should be matched with vacant energy in the normal region. Hence at $V = \Delta$ we have a spike in this interplay effect. This can be seen explicitly from analyzing the terms and factors within Eq. (D.2a) and their dependence on energy.

The CVC observed in the ferromagnetic Josephson junction in Ref. [14] has been demonstrated to be exchange-field sensitive. If we assume the measured SGS to be MAR-related then the system has to be in the thermalized regime following our results. At the same time our calculations, along with theory existing for other cases (ballistic transparent, diffusive with no relaxation etc) suggest that features representing lower MAR numbers $n$ are more pronounced than higher numbers. For example the features at $\Delta$ and $2\Delta/3$ are stronger than those at $\Delta/2, 2\Delta/5$ etc. However, analyzing the CVC on Fig. 4 of Ref. [14] we see a peak at $V \approx \Delta = 180\mu eV$ and another, split peak at $V \approx 60\mu eV$ which corresponds to $\Delta/3$. If this feature is to be explained by MAR then some sort of signal should also be seen at several higher threshold voltages, $2\Delta/3, \Delta/2, 2\Delta/5$ which are not seen in this experiment. The only other suggested explanation of the measured SGS is that it corresponds to a minigap in the junction. Indeed, $60\mu eV$ agrees with the minigap formula $E_g \approx 3.12\varepsilon_{\text{Th}}$ for an SNS junction of the same dimensions [23, 24]. However, a minigap requires a strong, unsuppressed proximity effect. In particular, the minigap is quickly suppressed by low transparency interfaces, as well as by magnetic effects. The critical current in experiment Ref. [14] is strongly suppressed (as compared to a non-magnetic junction of the same geometry) indicating a weakened proximity effect. In this regime there should be absolutely no minigap in the system. Therefore, the nature of the SGS and its exchange-sensitive peak observed in Ref. [14] remains a mystery.

# 6 Conclusion

To conclude we have calculated $I(V)$ in long diffusive SINIS and SIFIS junctions with strong thermalization at intermediate temperatures, $\varepsilon_{\text{Th}} \ll T \ll \Delta$. We found a subharmonic gap structure which exhibits splitting in the presence of an exchange field $h_{\text{ex}}$, with the splitting proportional to the voltage: MAR-related features are seen at $V_{n\pm} = (\Delta \pm h_{\text{ex}})/n$. We have shown that strong thermalization is essential to the field-induced splitting and that no splitting would happen in junctions with weak energy relaxation. Another striking difference is the apparent suppression of even MAR series in the SGS by thermalization.

# Acknowledgements

We thank V. V. Ryazanov, Ya. V. Fominov, and I. V. Bobkova for valuable discussions. This work was supported by the Russian Science Foundation (Grant No. 19-72-00125) and the Basic research program of Higher School of Economics. We acknowledge support of this work by Rosatom.

# A    Effective electron temperature in weak link

Here we present the derivation of the asymptotic value of the effective electron temperature $T_e$. Adopting the formula for heat flow between phonons of the substrate and electrons in metal $P_{e-ph}$ from Ref. [20], and heat flow of electrons through SN-interface $P(V)$ from Ref. [19], we write heat balance equations in the following form:

$$2P(V) = P_{e-ph}, \tag{A.1a}$$

$$P_{e-ph} = \Sigma \mathcal{V} \left( T_S^5 - T_e^5 \right), \tag{A.1b}$$

$$
\begin{aligned}
P(V) = \frac{\Delta}{2R_N} \Bigg( \Delta \Bigg[ &\left\{ K_0\left(\frac{\Delta}{2\log 2T_e}\right) + K_2\left(\frac{\Delta}{2\log 2T_e}\right) \right\} \cosh\left(\frac{V}{4\log 2T_e}\right) \\
&- K_0\left(\frac{\Delta}{2\log 2T_S}\right) - K_2\left(\frac{\Delta}{2\log 2T_S}\right) \Bigg] - V \sinh\left(\frac{V}{4\log 2T_e}\right) K_1\left(\frac{\Delta}{2\log 2T_e}\right) \Bigg).
\end{aligned} \tag{A.1c}
$$

Here $\Sigma$ is a material-dependent coefficient, related to $\tau_{\mathrm{in}}$ as in Ref. [20], $\mathcal{V}$ is the volume of the normal region. We expect $T_e \approx T_S$, therefore Eq. (A.1) can be approximately solved under conditions, presented in Sec. 2. For $T_e \ll \Delta$ and $V < 2\Delta$ we obtain following relation:

$$T_e = T_S - \frac{1}{5\Sigma \mathcal{V} T_S^4} \frac{\Delta}{2e^2 R} \sqrt{\frac{\pi T_S \log 2}{\Delta}} \left( \Delta - \frac{V}{2} \right) e^{-\frac{\Delta}{2\log 2T_S}\left(1 - \frac{V}{2\Delta}\right)}. \tag{A.2}$$

The correction to the effective temperature, i.e. the last term in Eq. (A.2) is not small at $V > 2\Delta$. Therefore, this approach is not applicable at above-gap voltages. In thus regime, the distribution function is controlled by coupling to the superconducting leads, rather than the substrate and is not close to a thermal distribution.

# B    Electric potential

To calculate the approximation of the electric potential in the leading order, which is the order $r^{-2}$, we take trace of Keldysh component of Green's function and perform inverse Fourier transform and obtain following relation:

$$\varphi(t) = \frac{1}{2} \int d\varepsilon \left\{ h_3^{(2)} + \frac{1}{4}\left(g_2^R - g_1^R\right) \circ h_0^{(0)} + \frac{1}{4} h_0^{(0)} \circ \left(g_2^A - g_1^A\right) \right\}. \tag{B.1}$$

It is easy to see from the definition of $\varphi_-$, that time-independent terms cancel out. The remaining ones are exponentially suppressed away from the NS boundaries when $T_e/\varepsilon_{\mathrm{Th}} \gg 1$. Near the superconductor, these terms contain an additional smallness of order $\varkappa_{T_e/\varepsilon_{\mathrm{Th}}}^{-1}$, which comes from the definition of $\nu(\varepsilon)$, which appears in every order of $\varphi$. We conclude that in the limit $\varepsilon_{\mathrm{Th}} \ll T$ the electric potential can be neglected.

# C    Coefficients

Here we present explicit expression for the coefficients in Eq. (15), which are obtained from straightforward solution of system of equations (14). Here $\Delta(n,m)$ is Kronecker delta symbol.

$$A_n^{(2)}(\varepsilon) = -\frac{1}{8\varkappa_{nV}\, r\, \sin\left[\varkappa_{nV}/2\right]} \times$$
$$\left[\Delta(n,1)\left\{-\left[f_{S,+}^A\delta h^+ - f_{S,+}^R\delta h_{++}^-\right]v_-^R + \left[f_{S,-}^A\delta h_{--}^+ - f_{S,-}^R\delta h^-\right]v_+^A\right\} + \right.$$
$$\Delta(n,-1)\left\{-\left[f_{S,+}^R\delta h^+ - f_{S,+}^A\delta h_{++}^-\right]v_-^A + \left[f_{S,-}^R\delta h_{--}^+ - f_{S,-}^A\delta h^-\right]v_+^R\right\} +$$
$$\Delta(n,0)\left\{-\left[f_{S,-}^A\delta h_{--}^+ - f_{S,-}^R\delta h^-\right]u_-^R - \left[f_{S,-}^R\delta h_{--}^+ - f_{S,-}^A\delta h^-\right]u_-^A +\right.$$
$$\left.\left.\left[f_{S,+}^R\delta h^+ - f_{S,+}^A\delta h_{++}^-\right]u_+^R + \left[f_{S,+}^A\delta h^+ - f_{S,+}^R\delta h_{++}^-\right]u_+^A\right\}\right], \quad \text{(C.1a)}$$

$$B_n^{(2)}(\varepsilon) = \frac{1}{8\varkappa_{nV}\, r\, \cos\left[\varkappa_{nV}/2\right]} \times$$
$$\left[\Delta(n,1)\left\{\left[f_{S,+}^A\delta h^+ - f_{S,+}^R\delta h_{++}^-\right]v_-^R + \left[f_{S,-}^A\delta h_{1--} - f_{S,-}^R\delta h^-\right]v_+^A\right\} + \right.$$
$$\Delta(n,-1)\left\{\left[f_{S,+}^R\delta h^+ - f_{S,+}^A\delta h_{++}^-\right]v_-^A + \left[f_{S,-}^R\delta h_{--}^+ - f_{S,-}^A\delta h^-\right]v_+^R\right\} +$$
$$\Delta(n,0)\left\{\left[f_{S,-}^A\delta h_{--}^+ - f_{S,-}^R\delta h^-\right]u_-^R + \left[f_{S,-}^R\delta h_{--}^+ - f_{S,-}^A\delta h^-\right]u_-^A\right.$$
$$\left.\left.+ \left[f_{S,+}^R\delta h^+ - f_{S,+}^A\delta h_{++}^-\right]u_+^R + \left[f_{S,+}^A\delta h^+ - f_{S,+}^R\delta h_{++}^-\right]u_+^A\right\}\right]. \quad \text{(C.1b)}$$

From the form of coefficients $A_n^{(2)}, B_n^{(2)}$ we determine that for applicability of perturbation theory relation $h^{(0)} \gg h_{0,3}^{(2)}$ has to be satisfied. This translates to $\gamma r^2 \gg 1$.

## D Contributions to the total current

Below we present time-dependent contributions to the total current in the first order of $r^{-1}$, which are mentioned in Eq. (18). Here the term $J_{h_s}$ corresponds to a contribution, dependent on distribution function of the superconducting leads $h_{S,1}, h_{S,2}$:

$$J_{h_s}^{(1)} = \int d\varepsilon \tanh\left(\frac{\varepsilon}{2T_S}\right)\left(f_S^A(\varepsilon) - f_S^R(\varepsilon)\right) \times$$
$$\left[e^{2itV}v_{++}^A - v_{--}^A e^{-2itV} + e^{2itV}v_{--}^R - v_{++}^R e^{-2itV}\right], \quad \text{(D.1a)}$$

$$J_+^{(1)} = \int d\varepsilon \tanh\left(\frac{\varepsilon}{2T_e}\right)\left[v_+^A f_{S,-}^R - f_{S,+}^A v_-^R + f_{S,+++}^A v_+^A - f_{S,---}^R v_-^R\right], \quad \text{(D.1b)}$$

$$J_-^{(1)} = \int d\varepsilon \tanh\left(\frac{\varepsilon}{2T_e}\right)\left[-v_-^A f_{S,+}^R + f_{S,-}^A v_+^R - f_{S,---}^A v_-^A + f_{S,+++}^R v_+^R\right]. \quad \text{(D.1c)}$$

Next we present explicit expression of each contribution to the second order correction to the current $I^{(2)}$ (see Eq. (20)).

$$J_h^{(2)} = \int d\varepsilon \left(g_{S,+}^R - g_{S,+}^A\right)\left(u_-^A\left(\delta h^- f_{S,-}^A - \delta h_{--}^+ f_{S,-}^R\right) + u_-^R\left(\delta h^- f_{S,-}^R - \delta h_{--}^+ f_{S,-}^A\right)\right) +$$
$$\left(g_{S,-}^A - g_{S,-}^R\right)\left(u_+^A\left(\delta h^+ f_{S,+}^A - \delta h_{++}^- f_{S,+}^R\right) + u_+^R\left(\delta h^+ f_{S,+}^R - \delta h_{++}^- f_{S,+}^A\right)\right), \quad \text{(D.2a)}$$

$$J_g^{(2)} = \frac{1}{2}\int d\varepsilon \left[ \left(g_{S,-}^R - g_{S,-}^A\right)\delta h^- \left(u_-^{R^2} + u_-^{A^2} + v_+^{R^2} + v_+^{A^2}\right) - \right.$$
$$\left. \left(g_{S,+}^R - g_{S,+}^A\right)\delta h^+ \left(u_+^{R^2} + u_+^{A^2} + v_-^{R^2} + v_-^{A^2}\right)\right], \quad \text{(D.2b)}$$

$$J_f^{(2)} = \frac{1}{2}\int d\varepsilon\; v_-^A v\left(\frac{V}{2}-\varepsilon\right)\left(g_{S,+}^A + g_{S,---}^A\right)\left(\delta h^- f_{S,-}^A - \delta h_{--}^+ f_{S,-}^R\right) +$$
$$v_-^R v\left(\varepsilon - \frac{V}{2}\right)\left(g_{S,+}^R + g_{S,---}^R\right)\left(-\delta h^- f_{S,-}^R + \delta h_{--}^+ f_{S,-}^A\right) -$$
$$2u_-^A g_{S,-}^A u\left(\frac{V}{2}-\varepsilon\right)\left(-\delta h^- f_{S,-}^A + \delta h_{--}^+ f_{S,-}^R\right) +$$
$$2u_-^R u\left(\varepsilon - \frac{V}{2}\right)g_{S,-}^R\left(-\delta h^- f_{S,-}^R + \delta h_{,--}^+ f_{S,-}^A\right) -$$
$$v_+^R v\left(\varepsilon + \frac{V}{2}\right)\left(g_{S,-}^R + g_{S,+++}^R\right)\left(-\delta h^+ f_{S,+}^R + \delta h_{++}^- f_{S,+}^A\right) +$$
$$v_+^A v\left(-\varepsilon - \frac{V}{2}\right)\left(g_{S,-}^A + g_{S,+++}^A\right)\left(-\delta h^+ f_{S,+}^A + \delta h_{++}^- f_{S,+}^R\right) -$$
$$2u_+^R u\left(\varepsilon + \frac{V}{2}\right)g_{S,+}^R\left(-\delta h^+ f_{S,+}^R + \delta h_{++}^- f_{S,+}^A\right) +$$
$$2u_+^A g_{S,+}^A u\left(-\varepsilon - \frac{V}{2}\right)\left(-\delta h^+ f_{S,+}^A + \delta h_{++}^- f_{S,+}^R\right). \quad \text{(D.2c)}$$

Here we also use the $\Phi_\pm = \Phi(\varepsilon \pm V/2)$, $\Phi_{\pm\pm} = \Phi(\varepsilon \pm V)$, $\Phi_{\pm\pm\pm} = \Phi(\varepsilon \pm 3V/2)$ notation.

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
