# Peer review of "Multiple Andreev reflections in diffusive SINIS and SIFIS junctions"

_SciPost Physics, doi:SciPost Phys. 15, 100 (2023)_

## Round 1 · Referee Report · Anonymous (Referee 1) · 2022-12-10

Report

The paper addresses the problem of multiple Andreev reflections in diffusive SINIS/SIFIS junctions in the limit of strong thermalization and weak SN tunneling -- which allows the authors to treat the problem perturbatively in tunneling, starting from the thermal-equilibrium state as the zeroth-order approximation.

I appreciate the good introduction, where the authors explain the background, the motivation and the principles of their approach. Also, I like the informative conclusion with an analysis of the results.

However, the main technical part was less understandable. While I didn't try to reproduce the full calculation, I would like to be able to clearly understand its main steps. And here I find the presentation in the paper somewhat confusing. Below I list some of my points of confusion that I would like to be resolved before I can recommend the paper for publication.

  1. The system involves several energy scales (E_Th, Delta, T, 1/tau_in), together with the tunneling probability 1/r. I would like the authors to clearly state the region of parameters, where their model is applicable. While I understand the main assumption E_Th << T << Delta, the conditions on tau_in and r are less clearly formulated and are scattered across different sections of the paper. I would like to have them all at the same place (either in the introduction, which may somewhat break the order of the presentation, or in the conclusion).

  2. In connection to the previous point, below equations (8), where the authors linearize the Usadel equations, they write that this linearization is only valid at small f, and, in the formula inside the text, they give an estimate of the corresponding small energy window around the gap edge. This looks strange: the energy window given by this estimate is very small, and it is not clear how this is compatible with the rest of the calculation (which, to my understanding, involves all the energies).

  3. I find notation very complicated and confusing. For example, equations (28) in Appendix D include objects like f^R_{S,+++} -- which I could not find defined anywhere. And this is not the only example of notation not clearly defined.

  4. Ideally, every notation (or at least the most important ones) should be introduced in words with some explanation of what it means: this would help the reader to understand the calculation beyond a sequence of formulas. For example, in Eq.(21) I see the function u(...) and I would appreciate to have a comment below this equation, something like " where u(...) is ... introduced in ...". I would guess it is the function u(epsilon) from Eq.(10c), but, since it was 3 pages above, it would be good to confirm to the reader that it is the right function (and, ideally, explain its meaning).

  5. The notation J(2)_\Sigma is only introduced in the caption to Fig.3 (at least as I could find it), and again, without any explanation. I also find it confusing that the figure is called "second-order contributions to the current" -- so I would expect that it would show different contributions from Eq.(20). Instead, it shows the first contribution only and then the total, without any explanation of this choice of plots. I find it confusing.

  6. The same about Fig.6. Its caption is even more confusing, because it refers to J(2)_\Sigma, without any description, so that the reader is supposed to find the definition in the caption to Fig.3.

  7. I appreciate very much the discussion of the results in Section 5, in particular mentioning the remaining unresolved questions. One of such questions is the explanation of the "even" series of features. I find it somewhat worrying that the authors do not have a full understanding of those features even in their own results. After all, their calculation is fully perturbative and controlled, and I would suppose that in this situation the origin of all the features should be crystal clear.

Finally, a couple of cosmetic comments that the authors may want to take into account in future revisions (but they are of minor importance):

  1. The scale of Fig.3 is very small, which makes it hardly readable.

  2. In many sentences, I would recommend to separate the introductory phrase by a comma, for better clarity. For example, in the second paragraph of Introduction there are two such sentences: "At voltages below the superconducting gap 2*Delta, electrons that ..." and "After a number of iterations, enough energy ..." There are a few other examples scattered across the paper.

To summarize, I would like to have points 1-6 clarified (and some response from the authors regarding p.7) before I can recommend the paper for publication.

---

## Round 1 · Referee Report · Anonymous (Referee 2) · 2023-2-27

Report

This is skillful and interesting contribution to the physics of nonequilibrium effects in Josephson junctions. The authors calculate I-V characteristics in long diffusive SINIS and SIFIS junctions. Strong thermalization regime is considered which allows to treat the weak link N (or F) region as a reservoir. Important result is the prediction that a subharmonic gap
structure exhibits splitting in the presence of an exchange field, with the splitting proportional to the voltage. At the same time, no splitting takes place in the case of weak energy relaxation. The developed approach is applied to analyze the experimental data from Golikova et al., Phys. Rev. B 86, 064416 (2012), where a subgap structure on I-V characteristics was observed in a Josephson junction with ferromagnetic interlayer, and it is concluded that the present model does not describe the observed effects.

To my opinion, this work deserves publication since it provides a solution of a complex theoretical problem within well-defined approximations and it may stimulate further studies in this direction. The manuscript is clearly written. Most of relevant literature is properly cited with exception of related work PRB 68, 224513 on nonequilibrium charge transport in double-barrier SINIS Josephson junctions. I can recommend the manuscript for publication if the authors add a comment on a relation of the present study and PRB 68, 224513

Requested changes

Add a reference to PRB 68, 224513 and discuss a relation to the present study

---

## Round 2 · Referee Report · Anonymous (Referee 2) · 2023-5-21

Report

I suggest publication of the resubmitted version of the manuscript

---

## Round 2 · Referee Report · Anonymous (Referee 1) · 2023-6-16

Report

The authors have satisfactorily addressed my comments and questions from the previous report, and I can now recommend the paper for publication.

However, I would like to point out to the authors that there is still room for improvement of their presentation. Namely, whenever they introduce a new notation, it should be accompanied by a few words explaining it, so that the reader does not have to guess.

For example, the authors introduce the temperature T_S in the first paragraph of Section 2. But it is only in the last paragraph of the same Section 2 that it is spelled out that this is the "lead temperature". It would have improved readability if it is explicitly said at the first mention of T_S.

Similarly, above Eq. (5a), they introduce a new notation G_{left,right}. It would help the reader if the authors explicitly mention that this is the Green's function at the SN interfaces.

Such explanations would improve readability, but I believe that an expert in the field would be able to reproduce the calculations already from the existing version of the manuscript. So I would leave it as an optional suggestion (and something the authors should pay attention to in their future manuscripts).

---

## Round 2 · Author Response

We thank the referees for their thoughtful reports and helpful comments. In our resubmitted manuscript, we have tried to address all the issues raised by the referees. We believe that the provided changes have improved the paper and also made it clearer for the reader. Detailed answers to the questions and comments of the referees follow below. Sincerely yours, Polkin A., Ioselevich P.

Replies to the questions of anonymous report 1

  1. The system involves several energy scales $(E_{Th},\ \Delta,\ T,\ 1/\tau_{in})$, together with the tunneling probability $1/r$. I would like the authors to clearly state the region of parameters, where their model is applicable. While I understand the main assumption $E_{Th} \ll T \ll \Delta$, the conditions on $\tau_{in}$ and $r$ are less clearly formulated and are scattered across different sections of the paper. I would like to have them all at the same place (either in the introduction, which may somewhat break the order of the presentation, or in the conclusion).

    There is one condition on $\tau_\mathrm{in}$, which allows the perturbative treatment of the distribution function with the zeroth approximation being a thermalized state. The condition is $\tau_{in} E_{Th} \ll r^2$. We added it together with a brief explanation to the first paragraph of the Sec. 2. Now all relevant conditions are collected in one place of the paper. 2. >In connection to the previous point, below equations (8), where the authors linearize the Usadel equations, they write that this linearization is only valid at small $f$, and, in the formula inside the text, they give an estimate of the corresponding small energy window around the gap edge. This looks strange: the energy window given by this estimate is very small, and it is not clear how this is compatible with the rest of the calculation (which, to my understanding, involves all the energies).

    Inequality below Eq. (8) contained a typo, there was a $\ll$ instead of $\gg$, which we now fixed. We corrected a similar typo in the caption of Fig. 2 (from $L\ll\sqrt(D/\Delta)$ to $L\gg\sqrt(D/\Delta)$). Physically, our theory is inapplicable in a small energy window around $\Delta$. This is addressed via regularization, where the particular regularization scheme does not affect our results. However, calculations become regularization-sensitive at higher orders of perturbation theory, which, fortunately, are outside the scope of this work.

  2. I find notation very complicated and confusing. For example, equations (28) in Appendix D include objects like $f^R_{S,+++}$ -- which I could not find defined anywhere. And this is not the only example of notation not clearly defined.}

    Indeed, to avoid exceedingly lengthy formulas we use an exceedingly large number of short notations. Functions with the subscript $F_{+++}$ are defined as a natural generalization of the notations $F_{+}=F(E+1/2 V)$ and $F_{++}=F(E+V)$, so that $F_{+++}(E) = F(E+3/2 V)$. There is one natural reason for such subscription stacking: whenever something is transferred between N and S, a $V/2$ change in energy occurs. Thus, pluses and minuses track the energy changes resulting from Andreev reflections. Looking at a series of $\pm$ subscripts across a product of multiple terms we can identify it with a particular MAR scheme. This is the easiest way of distinguishing the physical meaning of the many terms produced by the perturbation theory. We have added explanations to familiarize the reader with this notation (we also needed to distinguish these subscripts tracking energy accumulation from the order of perturbation theory in $1/r$ as well as from matrix indices, hence we decided to write $+,++,+++$ instead of another counter-type subscript such as $F_1,F_2,F_3$ etc.

    We expanded the definition after Eq. (11) and also placed a reminder of the notation after Eq. (29) in Appendix D.

  3. Ideally, every notation (or at least the most important ones) should be introduced in words with some explanation of what it means: this would help the reader to understand the calculation beyond a sequence of formulas. For example, in Eq.(21) I see the function u(...) and I would appreciate to have a comment below this equation, something like " where u(...) is ... introduced in ...". I would guess it is the function u(epsilon) from Eq.(10c), but, since it was 3 pages above, it would be good to confirm to the reader that it is the right function (and, ideally, explain its meaning).

    We added an explanation of the physical meaning of the functions $u(\dots)$ and $v(\dots)$ after Eq. (9): they correspond to incoherent and coherent propagators respectively. We have also added extensive explanations on the $F_{++\dots}$ notations (see prev. comment). We also added a reference to Eq. (10) after Eq. (21) and Eqs. (23).

    Furthermore, we have adjusted notations whereever possible, so that notations related to the same thing follow the same convention, e.g. order $i$ in perturbation theory in $r^{-1}$ is indicated as $F^{(i)}$, symbols related to $\pm V/2$ energy differences are $\pm$ etc.

  4. The notation $J^{(2)}_\Sigma$ is only introduced in the caption to Fig.3 (at least as I could find it), and again, without any explanation. I also find it confusing that the figure is called "second-order contributions to the current" -- so I would expect that it would show different contributions from Eq.(20). Instead, it shows the first contribution only and then the total, without any explanation of this choice of plots. I find it confusing.

    We changed the vertical axis label of Fig. 3.b, 3.c. We als swapped Figs 3 and 4. An explanation of why only the first current term is plotted on Fig. 4 is now found just below Fig. 4. Namely, only the first term is plotted because it is the only term containing features associated with Andreev reflections.

  5. The same about Fig.6. Its caption is even more confusing, because it refers to $J(2)_\Sigma$, without any description, so that the reader is supposed to find the definition in the caption to Fig. 3.

    We renamed the axes in Fig. 6. We changed the confusing notation and expanded the caption for this figure.

  6. I appreciate very much the discussion of the results in Section 5, in particular mentioning the remaining unresolved questions. One of such questions is the explanation of the "even" series of features. I find it somewhat worrying that the authors do not have a full understanding of those features even in their own results. After all, their calculation is fully perturbative and controlled, and I would suppose that in this situation the origin of all the features should be crystal clear.

    Thank you for pushing us to understand our own results more properly! After some thoughts we have come to a satisfactory interpretation of the even series. We have reworked the corresponding part of the Discussion to present it. In short, the even series are somewhat similar to even series in an energy-conserving setup. They are related to processes of AR-mediated transport where the involved particles are all below the gap (i.e. at $E < - \Delta$ with respect to the left superconductor). In other words, they correspond to MAR ladders that don't climb from below the left SC gap to above the right SC gap, but rather return to the left SC after an odd number of AR. This scheme yields the right order in $r^{-1}$ but cannot capture exchange-field induced splitting. To account for relaxation (which is necessary for splitting), we recast the process in terms of distribution functions and corrections from transport processes at NS interfaces.

  7. The scale of Fig. 3 is very small, which makes it hardly readable.

    We have increased the scale of the Figure. 9. >In many sentences, I would recommend to separate the introductory phrase by a comma, for better clarity. For example, in the second paragraph of Introduction there are two such sentences: "At voltages below the superconducting gap $2\Delta$, electrons that ..." and "After a number of iterations, enough energy ..." There are a few other examples scattered across the paper.

    We have added commas in the two sentences and in a number of others (around 10 in total)

Replies to the comment of anonymous report 2

The system that we considered in our work is indeed resembles the one considered in PRB 68, 224513. Some of our results (e.g. the first order contribution to current) can be directly compared with and do agree with the results of the referenced work. However, our calculations heavily rely on thermalization so that weak tunneling is required. Conversely, Brinkman et al rely on the adiabatic approximation so that $V\ll \Delta$ is required. Therefore, it is only possible to compare our results up to the first order of the small parameter $r^{-1}$. In the parametric region where both theories are valid, i.e. $V\ll \Delta$, and only first order in $r^{-1}$, our results agree. We added the corresponding discussion after Eq. (19). Note also that the referenced work neglects proximity effects in the normal region. In our paper we study the contribution of these effects to the current, since they are responsible for the emergence of the $J^{(2)}_h$ term.

---

## Round 2 · List of Changes

1. Reworked our understanding of "even" SGS features presented in the Discussion.
  2. In the first paragraph of Sec. 2 added condition on $\tau_\mathrm{in}$ and elaborated on its meaning.
  3. Corrected a sign typo in the caption of Fig. 1 (now $L\gg\sqrt{\hbar D/\Delta}$).
  4. Corrected a typo (wrong sign) in inequality below Eq.~(8).
  5. Above Eq. (10), the physical meaning of functions $u(…)$ and $v(…)$ is now discussed.
  6. Added a fuller explanation of our notations below eq. (11) for $F_{+++}$ and described their physical meaning as a tracker for transport/Andreev reflection events.
  7. Below Eq. (19) a discussion of our results in comparison to PRB 68, 224513 was added.

Minor and cosmetic changes

  1. Swapped Fig. 3 and Fig. 4.
  2. Enlarged scales of axes labels and ticks for Fig. 4 (previously known as Fig. 3)
  3. After Eq. (21) and Eqs. (23) a reference to equation (10) was added.
  4. Extended captions of Figs. 2-6.
  5. Removed the confusing notation $J_\Sigma^{(2)}$ from figures 4(b), 6(b) and 6(d)
  6. In Appendix A we added a discussion of Eq. (25).
  7. Added commas in a number of long sentences for better readability.
  8. Rewrote Eq. (29b) in terms of $\delta h$ to avoid confusing notation.
  9. Changed notation from $\delta h_{1,2}$ to $\delta h^{\pm}$

---

## Editorial Decision

published